# The Effects of Dairy and Plant-Based Liquid Components on Lutein Liberation in Spinach Smoothies

**DOI:** 10.3390/nu15030779

**Published:** 2023-02-02

**Authors:** Jan Neelissen, Per Leanderson, Lena Jonasson, Rosanna W. S. Chung

**Affiliations:** 1Division of Diagnostics and Specialist Medicine, Department of Health, Medicine and Caring Sciences, Linköping University, 581 85 Linköping, Sweden; 2Occupational and Environmental Medicine Center, 581 85 Linköping, Sweden; 3Division of Prevention, Rehabilitation and Community, Department of Health, Medicine and Caring Sciences, Linköping University, 581 85 Linköping, Sweden

**Keywords:** lutein, spinach, smoothie, cow’s milk, coconut milk, soymilk, liberation

## Abstract

Lutein is a dietary lipophilic compound with anti-inflammatory properties. We have previously shown that dairy fat can improve the lutein content in spinach smoothies. It is, however, unclear whether fat concentrations and fermentation status in dairy products affect lutein liberation in smoothies. Moreover, plant-based milks vary in fat, protein, and fiber content which may affect lutein dissolution. This study aimed to provide translatable information to consumers by comparing lutein liberation in spinach smoothies made with different dairy or plant-based liquids in domestic settings. The smoothies were digested in vitro, and liberated lutein was measured by high-performance liquid chromatography (HPLC). High-fat and medium-fat cow’s milk, as well as coconut milk with and without additives, were found to significantly improve lutein liberation by 36%, 30%, 25%, and 42%, respectively, compared to blending spinach with water alone. Adjustment models suggested that the effects of cow’s milk and coconut milk were derived from fat and protein, respectively. On the other hand, soymilk with and without additives showed significantly reduced lutein liberation by 40% and 61%, respectively. To summarize, only 4 out of 14 tested liquids increased lutein liberation in spinach smoothies. The results highlight the importance of testing food companions for lipophilic active ingredients.

## 1. Introduction

Lutein is a potent antioxidant that is abundant in dark leafy vegetables such as spinach and kale. It is well-acknowledged that its accumulation in human eyes is essential for eye health [1]. A growing body of research from experimental models indicates that lutein exerts anti-inflammatory effects by suppressing major inflammatory pathways [2,3,4]. There is also consistent epidemiological evidence supporting the beneficial role of increased lutein intake in a number of common diseases, including macular degeneration [5] and cardiovascular disease [4]. As our bodies cannot synthesize lutein, there is a great interest in methods to optimize lutein intake.

Lutein intake and lutein liberation are not necessarily correlated [5]. Liberation refers to the portion of nutrients that is released from the food matrix and becomes available for intestinal absorption. The liberated free lutein is vulnerable to oxygen and heat degradation during food preparation [6]. As lutein is lipophilic, it requires emulsifiers to dissolve in the aqueous environment in the intestine [7]. Micellarized lutein is less prone to degradation compared to non-micellarized lutein. We have recently reported that most of the domestic spinach preparation methods involving heat significantly reduced the levels of lutein released from spinach in both a cooking time-dependent and cooking temperature-dependent manner [8]. On the other hand, preparation methods without heat such as homogenizing raw spinach in water using a domestic blender led to a significantly higher release of lutein. In addition, mixing homogenized raw spinach with dairy cream (i.e., making a smoothie) resulted in an even higher release of lutein. It is unclear if other sources of dairy fat improve lutein dissolution to a similar extent.

Smoothie drinks are commonly used by consumers as a method to increase fruit and/or vegetable consumption. It is usually made by blending fruit and/or vegetables, with a liquid component. The most common liquids used in domestic settings can be classified into either dairy or plant-based products.

Dairy products such as cow’s milk contain natural micelles and are generally regarded as beneficial for the micellarization of lipophilic nutrients [9]. However, it is unclear whether factors such as fat concentrations and fermentation status affect the emulsifying property of dairy products. In addition, dairy-free plant-based milks are gaining popularity as alternate smoothie components in the consumer market. Plant-based milks are usually made from nuts, beans, or grains. These milks contain micelles, fibers, and/or protein. There is evidence that the co-consumption of lutein with certain pure plant-based fibers [10] or protein [11] reduces lutein bioaccessibility or bioavailability. Some of the plant-based milks also contain unknown amounts of stabilizers and/or emulsifiers. The effect of these common additives on lutein dissolution is largely unknown.

Food items available in the consumer market typically contain a mix of macronutrients and/or other potentially active ingredients. Although it is important to study pure nutritional ingredients to pinpoint the mechanism of action, this kind of information cannot be applicable to daily life. In order to provide translatable information to the consumers, real market products should be compared. In our previous study, a smoothie was found to be the optimal spinach consumption method for lutein liberation [8]. The present study aimed to identify liquid components with the potential to optimize lutein liberation in the smoothie. By using a well-established in vitro digestion model [12], we compared 14 commercially available dairy and plant-based milks in terms of their ability to liberate lutein in smoothies made under domestic conditions.

## 2. Materials and Methods

All chemicals and enzymes were acquired from Sigma-Aldrich, Stockholm, Sweden, unless otherwise stated. Fresh spinach leaves (*Spinacia oleracea*), dairy, and plant-based milks were purchased from a local supermarket (Appendix A). Cow’s milk and yogurts with different fat content were included to assess the effect of dairy fat content and fermentation status on lutein liberation. As it is common that plant-based milks contain additional stabilizers and/or emulsifiers which may also affect lutein dissolution, versions without such additives were also tested if locally and commercially available.

### 2.1. Preparation of Smoothie Samples

Smoothies were made by mimicking the composition of smoothies commonly prepared by the public. As smoothies are typically made by blending equal amounts of spinach leaves, plus watery fruits, and a milk-based liquid component according to domestic recipes, we diluted all liquid components by a minimum of half with Milli-Q water to account for the liquid derived from the missing watery fruits. For the more concentrated liquids such as whipping cream and pure coconut milk, higher dilution was made according to domestic recipes. The dilution factors for each liquid component with water are listed in Table 1. In brief, we added approximately 15 g of spinach leaves and 30 mL of diluted dairy or plant-based milk components to each smoothie. A water-only control (spinach + tap water) was included. In order to assess if any of the liquids contained lutein, a series of liquid-only samples were prepared by diluting the liquids to the same extent as in spinach-containing counterparts.

Following the addition of ingredients, each sample was blended for 30 s in a domestic blender for thorough homogenization (Andrew James Smoothie Maker 5060146063944, Andrew James UK Ltd., Seaham UK) and subsequently transferred to a new container. The blender was rinsed with 15 mL water, which was added to the homogenate, for full sample retrieval. Between samples, the blender was thoroughly cleaned to reduce contamination risk. Thirty milliliters of the homogenate was transferred to a separate 50 mL Falcon tube to carry out in vitro digestion.

### 2.2. In Vitro Digestion

In vitro digestion was designed to simulate human digestion along the gastrointestinal tract. The amount of lutein liberated from each smoothie and emulsified after the in vitro digestion represents the amount of lutein available for intestinal absorption. This will be termed “lutein liberation” from now on. The in vitro digestion procedure was adopted from Chung et.al. [8]. Briefly, 3 mL of porcine pepsin (in 0.1 M hydrochloric acid) at a concentration of 29,240 units/mL was added to all samples before incubation at 37 °C in a water bath for 1 h. The pH in each sample was adjusted to 2.0 by addition of 1 M hydrochloric acid to simulate the acidic environment in the stomach. After stomach digestion, the pH was adjusted to 5.3 using 1 M sodium hydrogen carbonate, and 8 mL of a solution containing 2 mg/mL porcine pancreatin and 12 mg/mL porcine bile extract (in 0.1 M sodium hydrogen carbonate) was added. Subsequently, the pH was adjusted to 7.5 using 1 M sodium hydroxide and all samples were incubated at 37 °C in a water bath for 2 h. Samples were mixed occasionally. Amylase was not used because it is not required for digestion of spinach [13]. The final volumes of digestate ranged between 42–47 mL. All variations of volumes between the samples due to pH adjustments were recorded and used for normalization to generate final lutein liberation levels. Smoothies made with dairy and plant-based products were digested on separate days. Following incubation, the digestates were stored at 4 °C overnight until extraction the following day.

### 2.3. Extraction of Lutein

In order to assess the liberation of lutein, the micellarized portions of the digestates were extracted as previously described [8]. The undigested spinach residues and the immiscible lipid layer were separated by three cycles of centrifugation. First, the digestates were centrifuged for 10 min at 5000× *g*. An amount of 1600 µL of the micellarized middle phase was transferred to a new tube. This was subsequently spun for 2 min at 25,000× *g* after which 1200 µL of the middle phase was transferred to another tube and spun at 25,000× *g* for 2 min. From the resulting sample, 400 µL of the micellarized phase was collected in duplicates for extraction. Pure lutein standards (1.25 nmol, 2.5 nmol, and 5 nmol) were extracted in parallel with each batch of digestate samples. In brief, 400 µL of 95% ethanol with 0.1% butylated hydroxytoluene was added to each digestate sample followed by 30 s of vortexing at full speed. Next, 600 µL of hexane was added and samples were vortexed vigorously for 5 min. All samples were centrifuged for 2 min at 14,000× *g*. 400 µL of the upper hexane phase was transferred to a new tube and dried under nitrogen gas. The dried samples were stored at −20 °C until quantification.

### 2.4. Quantification of Lutein

Lutein levels were determined using high-performance liquid chromatography (HPLC) analysis as described [4,8,14]. Each sample was reconstituted in 1000 µL methanol and acetonitrile (Merck, Stockholm, Sweden) in a 1:4 ratio. Subsequently, samples were subjected to water bath sonication for 5 min, shaken for 1 min at 2400 rpm, and centrifuged for 2 min at 14,000× *g*. During this process, samples were kept in dark as much as possible. The evaporation risk of the mobile phase was previously assessed. No significant concentration effect was found during this process. The HPLC analyses were performed using a PU 980 HPLC Pump (Jasco Inc., Tokyo, Japan), C18-Chromolith^®^ Performance RP-18 endcapped 100-4.6 HPLC column (Merck KGaA, Darmstadt, Germany), and Jasco MD-2010 Plus Multiwavelength Detector (Jasco Inc., Japan). Flow rate was set at 3.0 mL/min and 30 µL of samples were manually injected. The reproducibility of the manual injections was verified prior to the start of the experiment by repeatably injecting lutein standards. The detector identified signals at 450 nm and the data were interpreted using the Clarity software, version 2.6.5 (DataApex, Praha, Czech Republic). Concentrations of lutein in the digestates were calculated relative to the standard curve obtained from the standards.

### 2.5. Data Interpretations

“Lutein liberation” represents the total amount of lutein generated per gram of spinach used in the smoothie (µmol of lutein/gram of spinach).

“Improvement in lutein liberation” was calculated as follows:$$\frac{\left(Liberated\;lutein\;of\;smoothie\right)-\left(Liberated\;lutein\;of\;water-only\;control\right)}{\left(Liberated\;lutein\;of\;water-only\;control\right)\;}\times 100\%$$

“Improvement in lutein liberation per g of fat or protein or fiber” was calculated as follows:$$\frac{Improvement\;in\;lutein\;liberation\;of\;a\;particular\;liquid\;component}{gram\;of\;fat\;or\;protein\;or\;fiber\;in\;the\;corresponding\;smoothie}$$

### 2.6. Statistical Analyses

GraphPad Prism (v8.4.2) (GraphPad Software Inc., San Diego, CA, USA) and IBM SPSS statistics v.27.0. were used for statistical analyses. One-way ANOVA and Dunnett’s multiple comparison tests were used to compare between means of any liquid varieties and water-only controls. One-way ANOVA and post hoc comparison with Bonferroni correction were used for comparing the mean of each liquid variety with every other variety. Linear regression was performed for P for trend. Spearman’s rank correlation analysis was performed to study the relationship between either macronutrients and lutein liberation or between two types of macronutrients. ANCOVA multiple comparison tests were used to adjust for individual types of macronutrient levels and a post hoc comparison with Bonferroni correction was used to compare the estimated mean of each liquid variety with every other variety. Differences were considered significant when *p* < 0.05. All values are presented as mean ± 95% CI.

## 3. Results

### 3.1. The Effects of Dairy Products on Lutein Liberation in Spinach Smoothies

In order to compare different liquid components for their effects on lutein liberation, spinach smoothies were made using various liquid components based on common domestic recipes. Thereafter, the smoothies were digested in vitro, and lutein contents in the micellarized phase of the digestates were measured and expressed in terms of µmol of lutein per gram of spinach used per smoothie. The dairy and plant-based liquids were tested in two separate arms of experiments. The water-only controls from both arms provided comparable lutein liberation levels. The liquid component-only controls resulted in no measurable lutein levels (Appendix A).

Within the dairy category, three types of cow’s milk and three types of cow’s milk-derived yogurts with various fat contents as well as whipping cream were compared. The smoothies with high-fat cow’s milk and medium-fat cow’s milk were the only dairy liquids that showed significant effects on lutein liberation compared to water showing 36% and 30% increase, respectively, both *p* < 0.01 (Figure 1A and Appendix A) when multiple comparisons within the dairy category were conducted. The effect observed in cow’s milk products was shown to be fat-dependent (*P*_trend_ = 0.019). Amongst the three types of cow’s milk, high-fat cow’s milk resulted in significantly higher lutein liberation than its low-fat counterpart (36% vs. 21% increase, *p* < 0.05). On the other hand, the smoothies with whipping cream containing far higher fat content showed only a 9% increase.

In contrast to cow’s milk, none of the cow’s milk-derived yogurt products were found to significantly improve lutein liberation compared to water (Figure 1A). A weak fat-dependent trend was, however, observed amongst the yogurt products (*P*_trend_ = 0.046). Although smoothies made with low-fat cow’s milk and low-fat yogurt contained the same final fat content, low-fat cow’s milk resulted in significantly higher lutein levels (*p* < 0.01, Appendix A), as shown in Figure 1B. Similarly, high-fat cow’s milk resulted in significantly more liberated lutein compared to mild yogurt (*p* < 0.05) despite similar fat contents in the two smoothies.

### 3.2. The Effects of Plant-Based Products on Lutein Liberation in Spinach Smoothies

We further investigated four kinds of commonly used plant-based milks derived from either soybeans, almonds, oats, or coconuts. It is common that additives such as stabilizers or emulsifiers are added to plant-based milks. If available, the same type of milks with and without additives was tested in parallel; and if possible, the two versions were acquired from the same manufacturer (Table 1, Appendix A and Appendix A).

As shown in Figure 1C, coconut milk was the only product that significantly improved lutein liberation within the plant-based category. Smoothies with pure coconut milk showed the highest levels of improvement compared to water (42% increase, *p* < 0.001, Appendix A), followed by coconut milk with additives (25% increase, *p* < 0.05).

On the other hand, multiple comparison analyses showed that soymilk products with or without additives significantly reduced lutein levels compared to water by 40% and 61%, respectively (Appendix A). No significant changes in lutein liberation were observed when using oat milk or almond milk.

### 3.3. Comparing Improvers and Reducers

In summary, four types of liquids: high- and medium-fat cow’s milk, and coconut milk with and without additives, were found to improve lutein liberation in spinach smoothies. They will hereafter be referred to as “improvers”. Accordingly, the two types of soymilks that reduced lutein liberation will be referred to as “reducers”. As the water-only controls (spinach + water) from the dairy and plant-based liquid arms of the experiments showed comparable results, the data from the improvers and reducers were pooled to perform additional multiple comparisons (Figure 1D). Among improvers, the high-fat and medium-fat cow´s milk yielded significantly higher lutein levels compared to the coconut milk with additives (1.3- and 1.2-fold, respectively).

### 3.4. The Relationship between Macronutrients and Lutein Liberation

We further assessed the relationships, either between macronutrients (i.e., fat, protein, carbohydrate, and fiber content in smoothies) and lutein liberation per gram of spinach or between macronutrients (Table 2). Only the carbohydrate content in smoothies was significantly correlated to lutein liberation (*r* = 0.468, *p* < 0.001). Amongst the macronutrients, the carbohydrate content was highly correlated to the protein content (*r* = 0.888, *p* < 0.001, *n* = 11) and the fiber content (*r* = 1.000, *p* < 0.001, *n* = 3) but not to the fat content. The content of fat, protein, and fiber did not show any significant relationship. Further adjustment modeling for the effects of individual macronutrients was performed with only fat, protein, and fiber as covariates.

To shed light on the effects of macronutrients on lutein liberation, the improvement of lutein liberation per gram of each type of macronutrient was first modeled in ANOVA with post hoc comparison using Bonferroni correction. The result from one type of macronutrient was subsequently adjusted for the result from the other macronutrient in ANCOVA with post hoc comparison using Bonferroni correction. Only the improvers and the reducers were included in the models comparing fat and protein. Only five types of liquids (all plant-based) contained fibers. Modeling for the effects of fibers was performed on these fiber-containing liquids.

### 3.5. Effects of Fat on Lutein Liberation

Before the adjustment, the model showed that liquid types had significant effects on the improvement of lutein liberation per gram of fat (Model *r*^2^ = 0.952, *p* < 0.001, Table 3). The four improvers were all significantly different from the two reducers (all *p* < 0.001, Appendix A). Medium-fat milk and coconut milk with additives showed comparable high levels of improvement with 135% and 141% compared to water-only control, respectively. Amongst the reducers, both types of soymilks significantly reduced lutein liberation per gram of fat but they were not significantly different from each other (−148% with additives, −225% pure soymilk).

After the adjustment for protein, the effect of liquid types on lutein liberation remained significant (Model *r*^2^ = 0.939, *p* < 0.001, Table 3), with medium-fat and high-fat cow’s milk showing the highest levels of improvement per gram of fat with 163% and 107% increase, respectively. Regarding the coconut milk products with or without additives, the adjusted model showed only 14% and 40%, respectively, of the improvement was contributed by the fat content. This model suggested that the effect of coconut milks was mainly due to its protein content (Table 3 and Appendix A).

### 3.6. Effects of Protein on Lutein Liberation

When considering the improvement in lutein liberation in smoothies based on protein content, the model indicated that six liquid types were significantly different (Model *r*^2^ = 0.921, *p* < 0.001, Table 4). After the adjustment for fat, these significant differences remained (Model *r*^2^ = 0.901, *p* < 0.001). The coconut milk with additives showed the largest improvement per gram of protein with a 1687% and 1249% increase before and after adjustment, respectively. Before adjustment, the pure coconut milk showed the second highest effect from protein (539%) while the cow’s milk varieties showed 7–9 times less improvement compared to the previous model, and the soymilk varieties even showed negative effects. After adjusting for fat, the estimated effects of all plant-based milks became positive while those from the dairy milks became negative. However, multiple comparisons of the protein effects indicated that only the coconut milk varieties differed significantly from the cow’s milk varieties (Appendix A). The two cow’s milks and the two soymilks were not different in terms of protein effect while the two coconut milks were significantly different from each other (*p* < 0.001). This model consolidated that the effect of coconut milks was probably mostly derived from protein. In addition, it also suggested that the effect of cow’s milks was probably not attributed to protein whereas the effect from soymilks was unclear. As soymilk contains fibers, its effect would be more accurately modeled if also taking the effect from fibers into account.

### 3.7. Effects of Fiber on Lutein Liberation

In order to further understand the origin of the negative effect observed in soymilk, its effect from fat was adjusted for the effect from both protein and fibers. All liquids that contained fat, protein, and fibers were included in this series of multiple comparisons. In the unadjusted model, both types of soymilks were significantly different from the other fiber-containing liquids with the pure soymilk variety showing the largest difference (all *p* < 0.001, Table 5 and Table 6). After adjusting for both protein and fibers, the effect of fat from both types of soymilks was abolished. This model suggested that the negative effect observed for the two types of soymilks was derived from both protein and fibers.

## 4. Discussion

This study comprehensively compared the most common dairy and plant-based liquid components for their abilities to liberate lutein in spinach smoothies produced according to domestic recipes and under domestic settings. As food components in the consumer markets contain varying combinations of macronutrients and/or additives, it is important to compare these products for appropriate advice to consumers. Our data clearly demonstrated that the lutein liberation capacity varied amongst the dairy and plant-based categories. Only 4 out of 14 commercially available liquids could optimize lutein levels in smoothies. Out of dairy-based products, those that were fermented or artificially concentrated did not perform as well as the relatively less processed cow’s milk varieties. In addition, the use of high- and medium-fat cow’s milk as well as coconut milks, regardless of additive content, resulted in significantly higher levels of liberated lutein compared to water-only controls, while soymilk variants showed opposite effects.

Consuming “green” smoothies has become a popular way for people to meet the recommended daily intake of vitamins and antioxidants. Many of these dietary compounds such as lutein are lipophilic and would require to be emulsified before absorption. As lutein is stored within the leaf matrix, its release from the matrix as well as its micellarization are believed to be crucial for its absorption (6). We recently reported that turning spinach into smoothies, i.e., mechanically homogenizing spinach leaves in a blender followed by the addition of dairy fat, was an effective domestic method to improve lutein levels [8].

Although it is a general fact that dairy fat improves micelle formation and carotenoid uptake [9,15], the present results indicate that not all dairy fat-containing products emulsify lutein to a similar extent. Fat dependency was observed in both cow’s milk and cow’s milk-derived yogurt categories. The emulsifying potential of cow’s milk products was, however, superior. Cow’s milk contains milk fat globules (MFGs) as natural emulsifiers. It is postulated that as fat concentration increases, the MFG content increases and so does the emulsifying potential of the milk product. However, our adjustment models suggested that the fat dependency can reach a plateau as the medium-fat cow’s milk showed higher levels of improvement per gram of fat compared to its high-fat counterpart. A similar plateau effect on micellarized levels of lutein was previously described by comparing whole (3.6% fat content) and semi-skimmed milk (1.55% fat content) in a similar in vitro digestion model [16]. It is also worth noting that low-fat cow’s milk was able to emulsify more lutein than high-fat yogurt in the present study, highlighting that the fermentation status of dairy products matters in terms of fat liberation. This phenomenon may be explained by differences in the morphology of MFGs. During the fermentation process, MFGs in the yogurt become aggregated leading to compromised emulsifying properties [17]. Similarly, whipping cream contains extremely high levels of MFGs which are likely to aggregate in high concentrations leading to reduced emulsifying potential. However, Xavier et al. found no difference in terms of micellarization of lutein between milks and their laboratory-made yogurts in matching and varying fat concentrations [16]. This inconsistency may indicate that the type of culture used as well as the fermentation method are the main determinants for the ability of fermented milk products to facilitate lutein dissolution.

Amongst the plant-based milks, only coconut milk was found to significantly increase lutein liberation compared to water. Coconut milk is made from homogenizing coconut meat with or without coconut water. Coconut meat contains coconut oil and high levels of coconut protein. Co-administration of coconut fat has been found to significantly increase plasma and tissue levels of lutein in mice fed with high-dose pure lutein compared to other plant-based oil [18]. The effect of co-consuming coconut protein and carotenoids has not been previously studied. However, coconut protein is known to play a major role in maintaining coconut milk as an emulsion [19]. It is thus possible that coconut protein as well as coconut fat promoted the emulsification of lutein in the spinach smoothies. This postulation was supported by our multiple comparison models showing that the improvement of lutein liberation in coconut milks-based smoothies was largely derived from the protein content.

According to our findings, the additives in the coconut milk may contribute to the protein effect on lutein liberation. Currently, knowledge of the effects of different types of additives on lutein liberation is scarce. In addition, manufacturers are not required to declare the quantity of additives in these types of products, hence it is not possible to predict the effect of these additives even if their effects were well-understood. However, it is worth noting that other plant-based liquids in our study contained similar combinations of additives without showing any significant effect on liberation when compared to their pure plant-based counterparts. This supports the assumption that additives alone do not affect lutein liberation in a significant manner. Further research is however needed to further understand the effects of commonly used additives on lutein liberation.

In the present study, the two soymilk varieties were the only liquids that showed a reduction in lutein liberation when compared to water. Data also indicated that this effect may be due to the protein and fiber content in soymilk. The result is supported by Iddir et al. who showed that soy protein isolate reduced micellarized lutein levels from spinach in a similar in vitro digestion system [11]. Furthermore, it has been demonstrated in a number of experimental studies that soy protein has the potential to adversely affect bile acid function [20] and reduce dietary fat absorption in rats [21]. Mechanistic studies have postulated that soy protein absorbs bile acid which in turn results in a lower capacity to emulsify dietary lipids [22]. Hence, it can be speculated that the protein in soymilk interfered with the bile acid in our digestion model. Additionally, the effect of soymilk may be explained by the pectin content in soybeans. There is evidence from a human absorption study that dietary fibers, such as pectin, can reduce lutein absorption [10]. A future fat absorption study in humans could further elucidate the effects of soy liquid as a smoothie component.

This study has some limitations. First, the digested smoothie samples were stored at 4 °C overnight before lutein was extracted the following day. Although lutein could be subjected to degradation during storage, all samples were stored with minimum exposure to oxygen, light, and heat. A substantial amount of lutein could be measured from all of our smoothie samples, so the detection of lutein was not limited due to low levels. Secondly, evaporation risk exists during the reconstitution step of lutein isolates. However, all samples were managed in the same manner to ensure they were comparable. Thirdly, the current model did not account for intestinal movement of digestate, diet-stimulated bile acid excretion, or the processes of intestinal absorption which could affect nutrient absorption and bioavailability. A previous human absorption study has indicated that each type of carotenoid requires a certain amount of dietary fat to facilitate intestinal absorption and that the effect would reach a plateau once the required amount is achieved [23]. Therefore, it is possible that the types of milk that were less efficient, such as low-fat milk and yogurt, could still be effective in vivo. This postulation is further supported by a human bioavailability study showing increased lutein levels in serum after one dose of lutein-fortified fermented milk with medium fat (1.7 g of fat/100 mL). However, these authors did not include commercially available milk or yogurt in the comparison [24]. Therefore, an in vivo model should be conducted to consolidate the effect of the identified liquid components on lutein absorption. Thirdly, some of the products contained unknown amounts of ingredients, such as additives, which are impossible to adjust for. This problem, however, is faced by all consumers on a daily basis. In order to provide translatable information to the consumers, it is essential to compare market products that may not be as well controlled as lab-generated food samples. Another potential limitation is that the fat content in the four improvers was mostly composed of saturated fat which may be a concern for consumers who are under restrictions on dietary fat intake. The smoothies based on pure coconut milk contained the highest fat content (2.4 g/100 mL). However, this should not be a health concern since such a smoothie would only account for around 18% of the recommended daily saturated fat intake in a standard 2000 calorie daily diet [25].

## 5. Conclusions

In conclusion, the present study demonstrated a large variation in lutein liberation when different dairy and plant-based liquids were used to prepare spinach smoothies in domestic settings. Both cow’s milk and coconut milk showed beneficial effects and could be recommended to improve lutein liberation, while soymilk showed opposite effects. As lipophilic nutrients, such as lutein, require emulsifiers to be absorbed intestinally, the present results highlight the importance of comparing different food companions for the delivery of lipophilic compounds.

## Figures and Tables

**Figure 1 nutrients-15-00779-f001:**
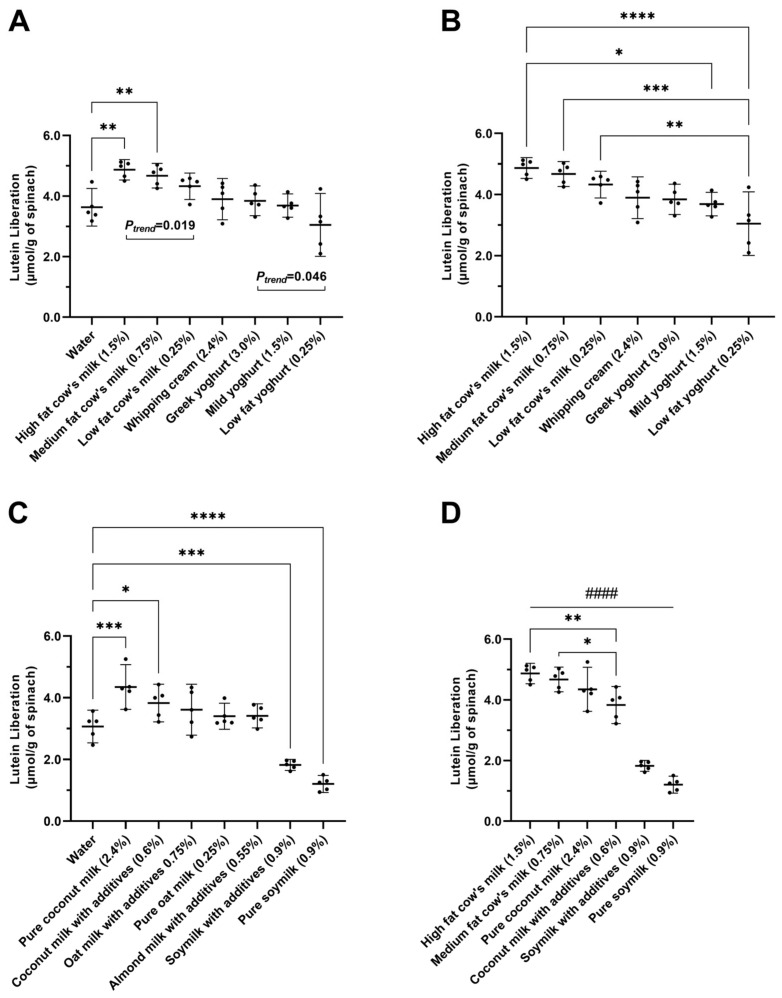
Levels of liberated lutein from spinach smoothies made with dairy or plant-based liquid components. Final fat concentrations in the resulting smoothies are listed in brackets. Data are presented in µmol of liberated lutein after in vitro digestion per gram of spinach used in smoothie. *n* = 5 for all groups. (**A**) Multiple comparisons of lutein liberation between smoothies with any dairy liquid and water are presented. P for trend (*P*_trend_) from linear regression amongst the milk or yogurt products is shown. (**B**) Multiple comparisons of lutein liberation amongst the smoothies made with dairy products (only the notable significances are shown). (**C**) Multiple comparisons of lutein liberation between smoothies with any plant-based liquids and water are presented. (**D**) Multiple comparisons of lutein liberation between the improvers and reducers * *p* < 0.05, ** *p* < 0.01, *** *p* < 0.001, and **** *p* < 0.0001 between two types of liquids. #### *p* < 0.0001 between any improvers and any reducers in Figure 1D. Error bars represent means with 95% CI.

**Table 1 nutrients-15-00779-t001:** Macronutrient and additive content of liquid components as well as macronutrient content in the corresponding smoothies.

Categories	Liquid Components	Brands	Product Fat (g/100 g) ^a^	Product Carbohydrate (g/100 g) ^a^	Product Protein (g/100 g) ^a^	Product Fiber (g/100 g) ^a^	Notable Product Additives ^b^	Dilution Factors ^c^	Smoothie Fat (g/100 mL) ^d^	Smoothie Carbohydrate (g/100 mL) ^d^	Smoothie Protein (g/100 mL) ^d^	Smoothie Fiber (g/100 mL) ^d^
Dairy	Low-fat yogurt	Arla ^f^	0.5	4.0	4.1			2	0.25	1.98	2.03	
Dairy	Mild yogurt	Arla ^f^	3.0	3.6	3.4			2	1.50	1.80	1.70	
Dairy	Greek yogurt	Arla ^f^	6.0	3.6	3.3			2	2.97	1.78	1.63	
Dairy	Low-fat cow’s milk	Arla ^f^	0.5	4.9	3.5			2	0.25	2.46	1.75	
Dairy	Medium-fat cow’s milk	Arla ^f^	1.5	4.9	3.5			2	0.75	2.46	1.76	
Dairy	High-fat cow’s milk	Arla ^f^	3.0	4.8	3.4			2	1.51	2.41	1.71	
Dairy	Whipping cream	ICA ^g^	36.0 ^e^	3.2 ^e^	2.3 ^e^		Stabilizer (carrageenan)	15	2.40	0.21	0.15	
Plant-based	Pure soymilk	Alpro ^h^	1.8 ^e^	2.3 ^e^	3.0 ^e^	0.5 ^e^		2	0.90	1.15	1.50	0.25
Plant-based	Soymilk with additives	Alpro ^h^	1.8 ^e^	0.0 ^e^	3.3 ^e^	0.6 ^e^	Acidity regulators (Potassium phosphates). Calcium carbonate. Stabilizer (gellan gum).	2	0.90	0	1.65	0.30
Plant-based	Pure oat milk	Oatly ^i^	0.5 ^e^	6.7 ^e^	1.0 ^e^	0.8 ^e^		2	0.25	3.35	0.50	0.40
Plant-based	Oat milk with additives	Oatly ^i^	1.5 ^e^	6.7 ^e^	1.0 ^e^	0.8 ^e^	Rapeseed oil. Calcium carbonate. Calcium phosphates. Potassium iodide.	2	0.75	3.35	0.50	0.40
Plant-based	Almond milk with additives	Alpro ^h^	1.1 ^e^	0.0 ^e^	0.4 ^e^	0.3 ^e^	Tri-calcium phosphate. Sea salt. Stabilizers (locust bean gum. gellan gum). Emulsifier (lecithins (sunflower)).	2	0.55	0	0.20	0.15
Plant-based	Pure coconut milk	Santa Maria ^j^	18.0	2.7	1.9			7.5	2.45	0.37	0.26	
Plant-based	Coconut milk with additives	Alpro ^h^	1.2 ^e^	0.0 ^e^	0.1 ^e^		Tri-calcium phosphate. Stabilizers (guar gum. xanthan gum. gellan gum). Sea salt.	2	0.6	0	0.05	

^a^ Macronutrient content based on product labeling. ^b^ Product additives that may affect emulsification. Concentrations are not available. Full list of product ingredients in Appendix A. ^c^ Dilution factors of liquid components by deionized water during the smoothie preparation. The dilution factor of 2 represents 15 mL liquid component + 15 mL deionized water. ^d^ Final macronutrient content in smoothie samples. ^e^ Concentration of macronutrients in g/100 mL. ^f^ Abbreviation for Arla Foods AB. ^g^ Abbreviation for ICA Gruppen AB. ^h^ Alpro is a part of Danone AB. ^i^ Abbreviation for Oatly Group AB. ^j^ Abbreviation for Santa Maria AB.2.2. In vitro digestion.

**Table 2 nutrients-15-00779-t002:** Correlations between lutein liberation and macronutrient content.

	Fat Content	Protein Content	Carbohydrate Content	Fiber Content
Lutein liberation/g of spinach	−0.004 (ns) (*n* = 14)	0.165 (ns) (*n* = 14)	0.468 (<0.001) (*n* = 11)	0.302 (ns) (*n* = 5)
Fat content		−0.187 (ns) (*n* = 14)	−0.171 (ns) (*n* = 11)	−0.263 (ns) (*n* = 5)
Protein content			0.888 (<0.001) (*n* = 11)	0.158 (ns) (*n* = 5)
Carbohydrate content				1.000 (<0.001) (*n* = 3)

Each cell indicates correlation coefficient (*p*-value). “ns” represents not significant.

**Table 3 nutrients-15-00779-t003:** Improvement in lutein liberation per gram of fat before and after adjustment for protein content.

	Unadjusted Model (*r*^2^ = 0.952, *p* < 0.001)			Adjusted Model (*r*^2^ = 0.939, *p* < 0.001)		
Liquid Components	Mean (%)	Confidence Interval		Estimated Mean (%)	Confidence Interval	
		Lower	Upper		Lower	Upper
Medium-fat cow’s milk	135.0	36.7	233.3	163.4	124.7	202.1
High-fat cow´s milk	79.6	25.4	133.8	106.8	68.5	145.1
Coconut milk with additives	140.6	77.1	204.0	14.0	−74.2	102.2
Pure coconut milk	57.2	44.1	70.3	39.8	3.9	75.7
Soymilk with additives	−148.3	−172.1	−124.6	−106.7	−150.1	−63.4
Pure soymilk	−225.4	−235.2	−215.5	−178.6	−2424.0	−133.2

Means and estimated means of improvement in lutein liberation associated with a particular type of liquid in spinach smoothies before and after adjusting for improvement in lutein liberation per gram of protein in an ANCOVA model.

**Table 4 nutrients-15-00779-t004:** Improvement in lutein liberation per gram of protein before and after adjustment for fat.

	Unadjusted Model (*r*^2^ = 0.921, *p* < 0.001)			Adjusted Model (*r*^2^ = 0.901, *p* < 0.001)		
Liquid Components	Mean (%)	Confidence Interval		Estimated Mean (%)	Confidence Interval	
		Lower	Upper		Lower	Upper
Medium-fat cow’s milk	57.5	15.6	99.4	−362.0	−697.5	−26.4
High-fat cow´s milk	70.3	22.4	118.1	−168.4	−420.3	83.4
Coconut milk with additives	1686.6	925.4	2447.9	1248.9	903.9	1593.9
Pure coconut milk	539.0	416.0	662.1	373.4	147.0	599.7
Soymilk with additives	−80.9	−93.9	−67.9	424.2	43.0	805.4
Pure soymilk	−135.2	−141.1	−129.3	621.3	95.6	1147.0

The unadjusted model was generated by ANOVA multiple comparisons with Bonferroni correction; The adjusted model was generated by ANCOVA multiple comparisons with Bonferroni correction using the effect of fat as a co-variant.

**Table 5 nutrients-15-00779-t005:** Improvement in lutein liberation per gram of fat before and after adjustment for protein and fibers.

	Unadjusted Model (*r*^2^ = 0.708, *p* < 0.001)			Adjusted Model (*r*^2^ = 0.856, *p* < 0.001)		
Liquid Components	Mean (%)	Confidence Interval		Estimated Mean (%)	Confidence Interval	
		Lower	Upper		Lower	Upper
Oat milk with additives	77.5	24.9	130.0	47.5	−223.1	318.1
Pure Oat milk	162.1	−79.0	403.3	−18.8	−307.9	270.4
Almond milk with additives	74.1	−6.1	154.3	158.5	−114.6	431.6
Soymilk with additives	−148.3	−172.1	−124.6	−82.8	−293.2	127.5
Pure soymilk	−225.4	−235.2	−215.5	−164.3	−777.6	449.0

The unadjusted model was generated by ANOVA multiple comparisons with Bonferroni correction. The adjusted model was generated by ANCOVA multiple comparisons with Bonferroni correction using the effect of protein and fibers as co-variant.

**Table 6 nutrients-15-00779-t006:** Multiple comparisons between liquid types in terms of improvement in lutein liberation per gram of fat before and after adjustment for protein and fibers.

	Unadjusted Model				Adjusted Model			
	Pure Oat Milk	Almond Milk with Additives	Soymilk with Additives	Pure Soymilk	Pure Oat Milk	Almond Milk with Additives	Soymilk with Additives	Pure Soymilk
Oat milk with additives	ns	ns	0.011	<0.001	ns	ns	ns	ns
Pure Oat milk		ns	<0.001	<0.001		0.012	ns	ns
Almond milk with additives			0.013	<0.001			ns	ns
Soymilk with additives				ns				ns

The unadjusted model was generated by ANOVA multiple comparisons with Bonferroni correction. The adjusted model was generated by ANCOVA multiple comparisons with Bonferroni correction using the effect of protein and fibers as co-variant. Each cell indicates *p*-value between two types of liquids. “ns” represents not significant.

## Data Availability

The data presented in this study are available on request from the corresponding author.

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
