# Peer review of "The Effects of Dairy and Plant-Based Liquid Components on Lutein Liberation in Spinach Smoothies"

_nutrients, 2023, doi:10.3390/nu15030779_

Round 1

Reviewer 1 Report

This manuscript shows the results of an interesting study on the release of lutein from various

types of frequently used beverages. Although the design, the methodology and the presentation of the results are adequate, the discussion is quite limited, since it does not include comparisons with other similar studies in the bibliography.

Major points: 1. Introduction: information from other studies on bioaccessibility / bioavailability of lutein from various foods or food supplements should be included. It is not enough to refer to their previous studies on this topic.  2. Lines 143 – 145: this volume (1 ml) of methanol: acetonitrile (1:4) seems easy to suffer concentration after being subjected to sonication for 5 min and vortexed 1 min and centrifuged 2 min. Did you used internal standard?. Please comment on that potential sample concentration.  3. Discussion should be rewritten to include comparisons with other studies on the bioaccessibility / bioavailability of lutein and/or other carotenoids from different types of foods. For example, studies on lutein enriched fermented milks and on the effect of modifiers on the carotenoid bioavailability from fruit juices (e.g., Granado-Lorencio, F et al. Lutein bioavailability from lutein ester-fortified fermented milk: in vivo and in vitro study. J. Nutr. Biochem., 21: 133-139; 2010 and, Granado-Lorencio, F et al. Bioavailability of carotenoids and a-tocopherol from fruit juices in the presence of absorption modifiers: in vitro and in vivo assessment. Br. J. Nutr. 101(4): 576-582; 2009) among other articles can be discussed. This review may also be of interest for the introduction and / or discussion: Böhm V et al. From carotenoid intake to carotenoid blood and tissue concentrations –

implications for dietary intake recommendations. Nutrition Reviews, 79(5): 544-573; 2021

Minor points: Table 3, p. 6,in the 5th column, the figure 14.0 is a typing mistake.  

Author Response

Comments of Reviewer 1:

Major points: 

  1. Introduction: information from other studies on bioaccessibility / bioavailability of lutein from various foods or food supplements should be included. It is not enough to refer to their previous studies on this topic.
  1. Lines 143 – 145: this volume (1 ml) of methanol: acetonitrile (1:4) seems easy to suffer concentration after being subjected to sonication for 5 min and vortexed 1 min and centrifuged 2 min. Did you used internal standard?. Please comment on that potential sample concentration.

  2. Discussion should be rewritten to include comparisons with other studies on the bioaccessibility / bioavailability of lutein and/or other carotenoids from different types of foods. For example, studies on lutein enriched fermented milks and on the effect of modifiers on the carotenoid bioavailability from fruit juices (e.g., Granado-Lorencio, F et al. Lutein bioavailability from lutein ester-fortified fermented milk: in vivo and in vitro study. J. Nutr. Biochem., 21: 133-139; 2010 and, Granado-Lorencio, F et al. Bioavailability of carotenoids and a-tocopherol from fruit juices in the presence of absorption modifiers: in vitroand in vivo assessment. Br. J. Nutr. 101(4): 576-582; 2009) among other articles can be discussed.This review may also be of interest for the introduction and / or discussion: Böhm V et al. From carotenoid intake to carotenoid blood and tissue concentrations –implications for dietary intake recommendations. Nutrition Reviews, 79(5): 544-573; 2021

Minor points: 

4. Table 3, p. 6,in the 5th column, the figure 14.0 is a typing mistake.  

Responses to Reviewer 1:

  1. Introduction: Thank you very much for the comment. References of other studies are added.
  2. Regarding the use of methanol:acetonitrile to reconstitute isolated lutein, the sonication steps were performed in a large water bath so the risk of evaporation was minimized. We have repetitively assessed the evaporation risk and found no observable difference in volume before and after reconstitution. Internal standard was not used as both the procedures and personnel were tested repetitively to ensure reproducibility. Also, all samples were processed the same way and should be comparable to one another.
    The use of a water-bath sonicator is now added to Section 2.4. Our consideration on evaporation risk is also mentioned in both Method and Discussion.
  3. Thank you very much for your comments regarding the discussion. In addition to all of the articles suggested, I have also added other studies to compare the results with our present data.
  4. Table 3. The figure 14.0 is not a typing mistake. The figure 14.0 indicates the effect of protein after adjustment. The fat content in coconut milk with additives contributed to only 14% of the improvement observed.

Reviewer 2 Report

Dear authors,

Thank you for submitting this interesting research investigating lutein liberation from various dairy and plant-based liquids. The findings are of interest.

I make a few minor suggestions for your consideration and revision of the manuscript:

- there are instances where the English grammatical language needs improvement, and I noticed these at times where plural and singular cases were mis-placed, for example in line 50, "Drinking smoothie is a popular....."replace with "Smoothie drinks are commonly used by consumers as a method to increase fruit and/ or vegetable consumption........" to make for a smoother English flowing language. 

- there are too many tables reported in the latter part of the results, and I feel the main message and presentation of research would be better served, by making these latter tables with the various adjusted and unadjusted models as supplementary tables (in particular from tables 4-8).

- some of the data in the tables are difficult to read and discern where it relates to various column headings, this could be improved with vertical line spacing, or shading, for easier reading between columns; also ensure the table heading remains with the table in the final pdf of the manuscript

- in the introduction there is reference to improvement in aged related macular degeneration (AMD) and CVD, and yet only CVD reference appears to be used (3). Also refer to relevant literature from AMD research.  There are only 12 references, so there is certainly space to include more references. 

Author Response

Comments of Reviewer 2:

  1. There are instances where the English grammatical language needs improvement, and I noticed these at times where plural and singular cases were mis-placed, for example in line 50, "Drinking smoothie is a popular....."replace with "Smoothie drinks are commonly used by consumers as a method to increase fruit and/ or vegetable consumption........" to make for a smoother English flowing language. 
  2. There are too many tables reported in the latter part of the results, and I feel the main message and presentation of research would be better served, by making these latter tables with the various adjusted and unadjusted models as supplementary tables (in particular from tables 4-8).
  3. Some of the data in the tables are difficult to read and discern where it relates to various column headings, this could be improved with vertical line spacing, or shading, for easier reading between columns; also ensure the table heading remains with the table in the final pdf of the manuscript
  4. In the introduction there is reference to improvement in aged related macular degeneration (AMD) and CVD, and yet only CVD reference appears to be used (3). Also refer to relevant literature from AMD research.  There are only 12 references, so there is certainly space to include more references. 

Responses to reviewer 2

  1. Thank you very much for the comment. The sentence is changed according to reviewer’s suggestion. Grammar is checked throughout the article.
  2. Table 4 and Table 6 are moved to the Supplementary section. The changes of tables are as follows:
    Table 4 --> Supplementary Table 5
    Table 5 --> Table 4
    Table 6 --> Supplementary Table 6
    Table 7 --> Table 5
    Table 8 --> Table 6

    Table 4 (now Suppl. Table 5) and Table 6 (now Suppl. Table 6) are moved to the Supplementary section. However, we believe that both Table 7 and Table 8 (now Table 5 and 6) are needed in the main text to fully interpret the results.

    The original supplementary materials were not included in the version of manuscript that the publisher asked me to edit. I have uploaded a new file with the original supplementary materials as well as the two new supplementary tables.

  3. Shading and vertical lines are added in Tables in the main text as well as in the Supplementary section to improve readability.
  4. Thank you very much for the comment. Additional references are added.